# RUBi: Reducing Unimodal Biases
# for Visual Question Answering

**Remi Cadene** [1]*, **Corentin Dancette** [1]*, **Hedi Ben-younes** [1], **Matthieu Cord** [1], **Devi Parikh** [2,3]

[1] Sorbonne Université, CNRS, LIP6, 4 place Jussieu, 75005 Paris,
[2] Facebook AI Research, [3] Georgia Institute of Technology
{remi.cadene, corentin.dancette, hedi.ben-younes, matthieu.cord}@lip6.fr,
parkih@gatech.edu

## Abstract

Visual Question Answering (VQA) is the task of answering questions about an image. Some VQA models often exploit unimodal biases to provide the correct answer without using the image information. As a result, they suffer from a huge drop in performance when evaluated on data outside their training set distribution. This critical issue makes them unsuitable for real-world settings.

We propose RUBi, a new learning strategy to reduce biases in any VQA model. It reduces the importance of the most biased examples, i.e. examples that can be correctly classified without looking at the image. It implicitly forces the VQA model to use the two input modalities instead of relying on statistical regularities between the question and the answer. We leverage a question-only model that captures the language biases by identifying when these unwanted regularities are used. It prevents the base VQA model from learning them by influencing its predictions. This leads to dynamically adjusting the loss in order to compensate for biases. We validate our contributions by surpassing the current state-of-the-art results on VQA-CP v2. This dataset is specifically designed to assess the robustness of VQA models when exposed to different question biases at test time than what was seen during training.

Our code is available: github.com/cdancette/rubi.bootstrap.pytorch

## 1 Introduction

The recent Deep Learning success in computer vision [1] and natural language understanding [2] allowed researchers to tackle multimodal tasks that combine visual and textual modalities [3, 4, 5, 6, 7]. Among these tasks, Visual Question Answering (VQA) attracts increasing attention. The goal of the VQA task is to answer a question about an image. It requires a high-level understanding of the visual scene and the question, but also to ground the textual concepts in the image and to use both modalities adequately. Solving the VQA task could have tremendous impacts on real-world applications such as aiding visually impaired users in understanding their physical and online surroundings, searching through large quantities of visual data via natural language interfaces, or even communicating with robots using more efficient and intuitive interfaces.

Several large real image VQA datasets have recently emerged [8, 9, 10, 11, 12, 13, 14]. Each one of them targets specific abilities that a VQA model would need to be used in real-world settings such as fine-grained recognition, object detection, counting, activity recognition, commonsense reasoning, etc. Current end-to-end VQA models [15, 16, 17, 18, 19, 20, 21, 22] achieve impressive

---

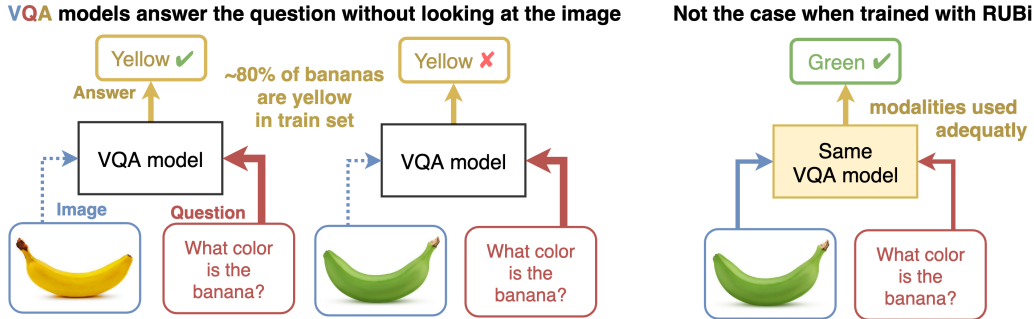

Figure 1: Our RUBi approach aims at reducing the amount of unimodal biases learned by a VQA model during training. As depicted, current VQA models often rely on unwanted statistical correlations between the question and the answer instead of using both modalities.

results on most of these benchmarks and are even able to surpass the human accuracy on a specific benchmark accounting for compositional reasoning [23]. However, it has been shown that they tend to exploit statistical regularities between answer occurrences and certain patterns in the question [24, 10, 25, 23, 13]. While they are designed to merge information from both modalities, in practice they often answer without considering the image modality. When most of the bananas are *yellow*, a model does not need to learn the correct behavior to reach a high accuracy for questions asking about the color of bananas. Instead of looking at the image, detecting a banana and assessing its color, it is much easier to learn from the statistical shortcut linking the words *what*, *color* and *bananas* with the most occurring answer *yellow*.

One way to quantify the amount of statistical shortcuts from each modality is to train unimodal models. For instance, a question-only model trained on the widely used VQA v2 dataset [9] predicts the correct answer approximately 44% of the time over the test set. VQA models are not discouraged to exploit these statistical shortcuts from the question modality, because their training set often follows the same distribution as their testing set. However, when evaluated on a test set that displays different statistical regularities, they usually suffer from a significant drop in accuracy [10, 25]. Unfortunately, these statistical regularities are hard to avoid when collecting real datasets. As illustrated in Figure 1, there is a crucial need to develop new strategies to reduce the amount of biases coming from the question modality in order to learn better behaviors.

We propose RUBi, a training strategy to reduce the amount of biases learned by VQA models. Our strategy reduces the importance of the most biased examples, i.e. examples that can be correctly classified without looking at the image modality. It implicitly forces the VQA model to use the two input modalities instead of relying on statistical regularities between the question and the answer. We take advantage of the fact that question-only models are by design biased towards the question modality. We add a question-only branch on top of a base VQA model during training only. This branch influences the VQA model, dynamically adjusting the loss to compensate for biases. As a result, the gradients backpropagated through the VQA model are reduced for the most biased examples and increased for the less biased. At the end of the training, we simply remove the question-only branch.

We run extensive experiments on VQA-CP v2 [10] and demonstrate the ability of RUBi to surpass current state-of-the-art results from a significant margin. This dataset has been specifically designed to assess the capacity of VQA models to be robust to biases by the question modality. We show that our RUBi learning framework provides gains when applied on several VQA architectures such as Stacked Attention Networks [26] and Top-Down Bottom-Up Attention [15]. We also show that RUBi is competitive on the standard VQA v2 dataset [9] when compared to approaches that reduce unimodal biases.

## 2 Related work

Real-world datasets display some form of inherent biases due to their collection process [27, 28, 29]. As a result, machine learning models tend to reflect these biases because they capture often undesirable

correlations between the inputs and the ground truth annotations [30, 31, 32]. Procedures exist to identify certain kinds of biases and to reduce them. For instance, some methods are focused on gender biases [33, 34], some others on the human reporting biases [35], and also on the shift in distribution between lab-curated data and real-world data [36]. In the language and vision context, some works evaluate unimodal baselines [37, 38] or leverage language priors [39]. In the following, we discuss about related works that assess and reduce unimodal biases learned by VQA models.

**Assessing unimodal biases in datasets and models**   Despite being designed to merge the two input modalities, it has been found that VQA models often rely on superficial correlations between inputs from one modality and the answers without considering the other modality [40, 32]. An interesting way to quantify the amount of unimodal biases that can potentially be learned by a VQA model consists in training models using only one of the two modalities [8, 9]. The question-only model is a particularly strong baseline because of the large amount of statistical regularities that can be leveraged from the question modality. With the RUBi learning strategy, we take advantage of this baseline model to prevent VQA models from learning question biases.

Unfortunately, biased models that exploit statistical shortcuts from one modality usually reach impressive accuracy on most of the current benchmarks. VQA-CP v2 and VQA-CP v1 [10] were recently introduced as diagnostic datasets containing different answer distributions for each question-type between train and test splits. Consequentially, models biased towards the question modality fail on these benchmarks. We use the more challenging VQA-CP v2 dataset extensively in order to show the ability of our approach to reduce the learning of biases coming from the question modality.

**Balancing datasets to avoid unimodal biases**   Once the unimodal biases have been identified, one method to overcome these biases is to create more balanced datasets. For instance, the synthetic datasets for VQA [23, 13] minimize question-conditional biases via rejection sampling within families of related questions to avoid simple shortcuts to the correct answer.

Doing rejection sampling in real VQA datasets is usually not possible due to the cost of annotations. Another solution is to collect complementary examples to increase the difficulty of the task. For instance, VQA v2 [9] has been introduced to weaken language priors in the VQA v1 dataset [8] by identifying complementary images. For a given VQA v1 question, VQA v2 also contains a similar image with a different answer to the same question. However, even with this additional balancing, statistical biases from the question remain and can be leveraged [10]. That is why we propose an approach to reduce unimodal biases during training. It is designed to learn unbiased models from biased datasets. Our learning strategy dynamically modifies the loss values to reduce biases from the question. By doing so, we reduce the importance of certain examples, similarly to the rejection sampling approach, while increasing the importance of complementary examples which are already in the training set.

**Architectures and learning strategies to reduce unimodal biases**   In parallel of these previous works on balancing datasets, an important effort has been carried out to design VQA models to overcome biases from datasets. [10] proposed a hand-designed architecture called Grounded VQA model (GVQA). It breaks the task of VQA down into a first step of locating and recognizing the visual regions needed to answer the question, and a second step of identifying the space of plausible answers based on a question-only branch. This approach requires training multiple sub-models separately. In contrast, our learning strategy is end-to-end. Their complex design is not straightforward to apply on different architectures while our approach is model-agnostic. While we rely on a question-only branch, we remove it at the end of the training.

The work most related to ours in terms of approach is [25]. The authors propose a learning strategy to overcome language priors in VQA models. They first introduce an adversary question-only branch. It takes as input the question encoding from the VQA model and produces a question-only loss. They use a gradient negation of this loss to discourage the question encoder to capture unwanted biases that could be exploited by the VQA model. They also propose a loss based on the difference of entropies between the VQA model and the question-only branch output distributions. These two losses are only backpropagated to the question encoder. In contrast, our learning strategy targets the full VQA model parameters to reduce the impact of unwanted biases more effectively. Instead of relying on these two additional losses, we use the question-only branch to dynamically adapt the value of the

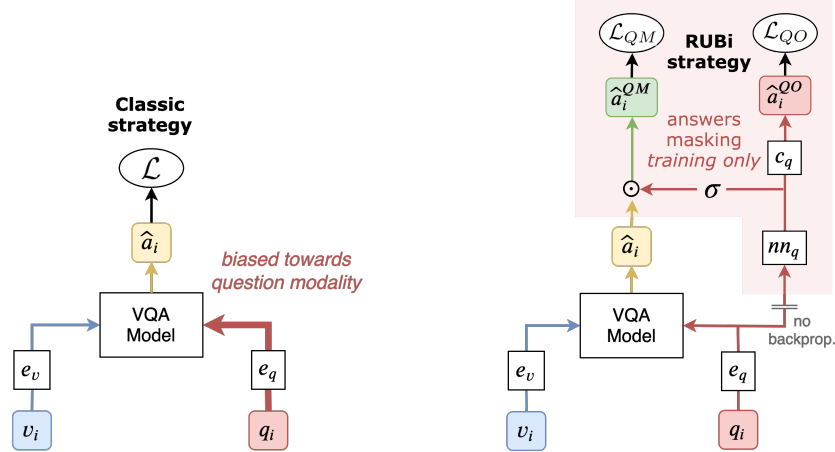

Figure 2: Visual comparison between the classical learning strategy of a VQA model and our RUBi learning strategy. The red highlighted modules are removed at the end of the training. The output $\hat{a}_i$ is used as the final prediction.

classification loss in order to reduce the learning of biases in the VQA model. A visual comparison between [25] and RUBi can be found in Figure 5 in the supplementary materials.

## 3 Reducing Unimodal Biases Approach

We consider the common formulation of the Visual Question Answering (VQA) task as a multi-class classification problem. Given a dataset $\mathcal{D}$ consisting of $n$ triplets $(v_i, q_i, a_i)_{i \in [1,n]}$ with $v_i \in \mathcal{V}$ an image, $q_i \in \mathcal{Q}$ a question in natural language and $a_i \in \mathcal{A}$ an answer, one must optimize the parameters $\theta$ of the function $f : \mathcal{V} \times \mathcal{Q} \to \mathbb{R}^{|\mathcal{A}|}$ to produce accurate predictions. For a single example, VQA models use an image encoder $e_v : \mathcal{V} \to \mathbb{R}^{n_v \times d_v}$ to output a set of $n_v$ vectors of dimension $d_v$, a question encoder $e_q : \mathcal{Q} \to \mathbb{R}^{n_q \times d_q}$ to output a set of $n_q$ vectors of dimension $d_q$, a multimodal fusion $m : \mathbb{R}^{n_v \times d_v} \times \mathbb{R}^{n_q \times d_q} \to \mathbb{R}^{d_m}$, and a classifier $c : \mathbb{R}^{d_m} \to \mathbb{R}^{|\mathcal{A}|}$. These functions are composed as follows:

$$f(v_i, q_i) = c(m(e_v(v_i), e_q(q_i)))$$
(1)

Each one of them can be defined to instantiate most of the state of the art models, such as [26, 41, 19, 42, 17, 43, 16] to cite a few.

**Classical learning strategy and pitfall** The classical learning strategy of VQA models, depicted in Figure 2, consists in minimizing the standard cross-entropy criterion over a dataset of size $n$.

$$\mathcal{L}(\theta; \mathcal{D}) = -\frac{1}{n} \sum_{i=1}^{n} \log(\text{softmax}(f(v_i, q_i)))[a_i]$$
(2)

VQA models are inclined to learn unimodal biases from the datasets [10]. This can be shown by evaluating models on datasets that have different distributions of answers for the test set, such as VQA-CP v2. In other words, they rely on statistical regularities from one modality to provide accurate predictions without having to consider the other modality. As an extreme example, strongly biased models towards the question modality always output *yellow* to the question *what color is the banana*. They do not learn to use the image information because there are too few examples in the dataset where the banana is not yellow. Once trained, their inability to use the two modalities adequately makes them inoperable on data coming from different distributions such as real-world data. Our contribution consists in modifying this cost function to avoid the learning of these biases.

### 3.1 RUBi learning strategy

**Capturing biases with a question-only branch** One way to measure the unimodal biases in VQA datasets is to train an unimodal model which takes only one of the two modalities as input. The key idea of our approach, depicted in Figure 2, is to adapt a question-only model as a branch of our VQA

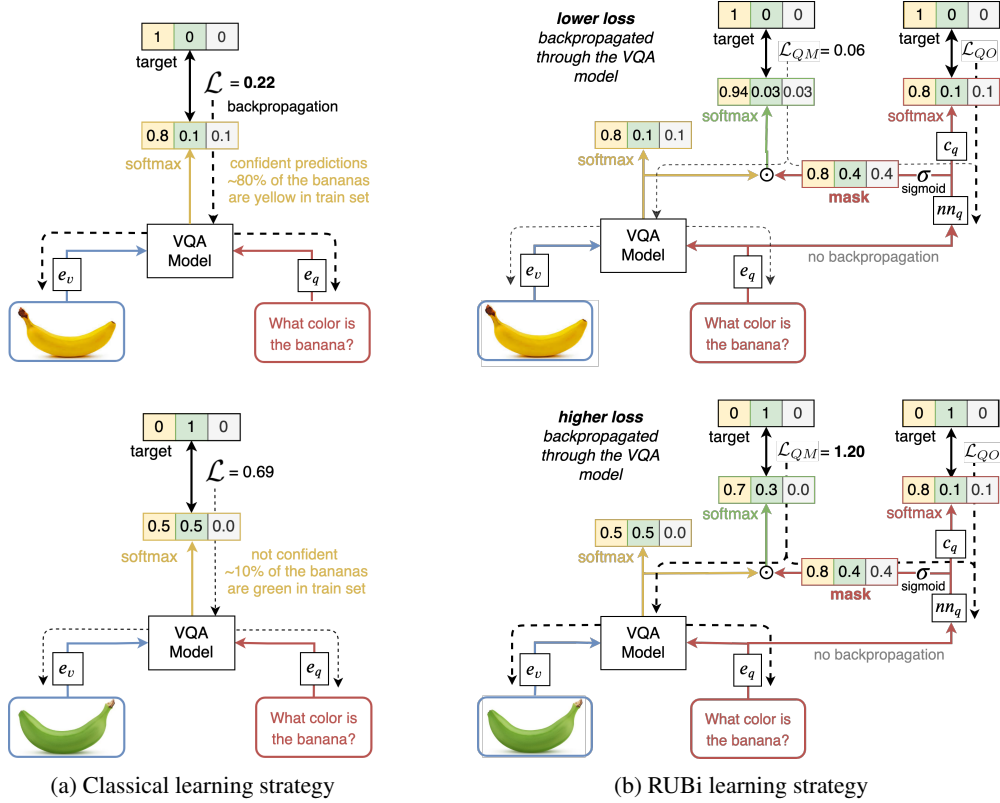

(a) Classical learning strategy

(b) RUBi learning strategy

Figure 3: Detailed illustration of the RUBi impact on the learning. In the first row, we illustrate how RUBi reduces the loss for examples that can be correctly answered without looking at the image. In the second row, we illustrate how RUBi increases the loss for examples that cannot be answered without using both modalities.

model, that will alter the main model's predictions. By doing so, the question-only branch captures the question biases, allowing the VQA model to focus on the examples that cannot be answered correctly using the question modality only. The question-only branch can be formalized as a function $f_Q : \mathcal{Q} \to \mathbb{R}^{|\mathcal{A}|}$ parameterized by $\theta_Q$, and composed of a question encoder $e_q : \mathcal{Q} \to \mathbb{R}^{n_q \times d_q}$ to output a set of $n_q$ vectors of dimension $d_q$, a neural network $nn_q \colon \mathbb{R}^{n_q \times d_q} \to \mathbb{R}^{|\mathcal{A}|}$ and a classifier $c_q \colon \mathbb{R}^{|\mathcal{A}|} \to \mathbb{R}^{|\mathcal{A}|}$.

$$f_Q(q_i) = c_q(nn_q(e_q(q_i))) \tag{3}$$

During training, the branch acts as a proxy preventing any VQA model of the form presented in Equation (1) from learning biases. At the end of the training, we simply remove the branch and use the predictions from the base VQA model.

**Preventing biases by masking predictions** Before passing the predictions of our base VQA model to the loss function defined in Equation (2), we merge them with a mask of length $|\mathcal{A}|$ containing a scalar value between 0 and 1 for each answer. This mask is obtained by passing the output of the neural network $nn_q$ through a sigmoid function $\sigma$. The goal of this mask is to dynamically alter the loss by modifying the predictions of the VQA model. To obtain the new predictions, we simply compute an element-wise product $\odot$ between the mask and the original predictions as defined in the following equation.

$$f_{QM}(v_i, q_i) = f(v_i, q_i) \odot \sigma(nn_q(e_q(q_i))))) \tag{4}$$

Our method modifies the predictions in this specific way to prevent the VQA model to learn biases from the question. To better understand the impact of our approach on the learning, we examine two scenarios. First, we reduce the importance of the most biased examples, i.e. examples that can be correctly classified without using the image modality. To do so, the question-only branch outputs

a mask to increase the score of the correct answer while decreasing the scores of the others. As a result, the loss is much lower for these biased examples. In other words, the gradients backpropagated through the VQA model are smaller, thereby reducing the importance of these examples in the learning. As illustrated in the first row of Figure 3, given the question *what color is the banana*, the mask takes a high value of 0.8 for the answer *yellow* which is the most likely answer for this question in the training set. On the other hand, the value for the other answers *green* and *white* are smaller. We see that the mask influences the VQA model to produce new predictions where the score associated with the answer *yellow* increases from 0.8 to 0.94. Compared to the classical learning approach, the loss is smaller with RUBi and decreases from 0.22 to 0.06. Secondly, we increase the importance of examples that cannot be answered without using both modalities. For these examples, the question-only branch outputs a mask that increases the score of the wrong answer. As a result, the loss is much higher and the VQA model is encouraged to learn from these examples. We illustrate this behavior in the second row of Figure 3 for the same question about the color of the banana. When the image contains a green banana, RUBi increases the loss from 0.69 to 1.20.

**Joint learning procedure**   We jointly optimize the parameters of the base VQA model and its question-only branch using the gradients computed from two losses. The main loss $\mathcal{L}_{QM}$ refers to the cross-entropy loss associated with the predictions of $f_{QM}(v_i, q_i)$ from Equation 4. We backpropagate this loss to optimize all the parameters $\theta_{QM}$ which contributed to this loss. $\theta_{QM}$ is the union of the parameters of the base VQA model, the encoders, and the neural network $nn_q$ of the question-only branch. In our setup, we share the parameters of the question encoder $e_q$ between the VQA model and the question-only branch. The question-only loss $\mathcal{L}_{QO}$ is a cross-entropy loss associated with the predictions of $f_Q(q_i)$ from Equation 3. We use this loss to only optimize $\theta_{QO}$, union of the parameters of $c_q$ and $nn_q$. By doing so, we further improve the question-only branch ability to capture biases. Note that we do not backpropagate this loss to the question encoder $e_q$ preventing it from directly learning question biases. We obtain our final loss $\mathcal{L}_{\text{RUBi}}$ by summing the two losses together in the following equation:

$$\mathcal{L}_{\text{RUBi}}(\theta_{QM}, \theta_{QO}; \mathcal{D}) = \mathcal{L}_{QM}(\theta_{QM}; \mathcal{D}) + \mathcal{L}_{QO}(\theta_{QO}; \mathcal{D}) \tag{5}$$

## 3.2   Baseline architecture

Most VQA architectures from the state of the art are compatible with our RUBi learning strategy. To test our strategy, we design a fast and simple architecture inspired from [16]. This baseline architecture is detailed in the supplementary material. As common in the state of the art, our baseline architecture encodes the image as a bag of $n_v$ visual features $\mathbf{v}_i \in \mathbb{R}^{d_v}$ using the pretrained Faster R-CNN by [15], and encodes the question as a vector $\mathbf{q} \in \mathbb{R}^{d_q}$ using a GRU, pretrained on the skipthought task [3]. The VQA model consists of a Bilinear BLOCK fusion [17] which merges the question representation $\mathbf{q}$ with the features $\mathbf{v}_i$ of each region of the image. The output is aggregated using a max pooling on the $n_v$ regions. The resulting vector is then fed into a MLP classifier which outputs the final predictions. While most of our experiments are done with this fast and simple baseline architecture, we experimentally demonstrate that the RUBi learning strategy is effective on other VQA architectures.

# 4   Experiments

**Experimental setup**   We train and evaluate our models on VQA-CP v2 [10]. This dataset was developed to evaluate the models robustness to question biases. We follow the same training and evaluation protocol as [25], who also propose a learning strategy to reduce biases. For each model, we report the standard VQA evaluation metric [8]. We also evaluate our models on the standard VQA v2 [9]. Further implementation details are included in the supplementary materials, as well as results on VQA-CP v1 and grounding experiments on VQA-HAT [44].

## 4.1   Results

**State-of-the-art comparison**   In Table 1, we compare our approach consisting of our baseline architecture trained with RUBi on VQA-CP v2 against the state of the art. To be fair, we only report approaches that use the strong visual features from [15]. We compute the average accuracy over 5 experiments with different random seeds. Our RUBi approach reaches an average overall accuracy

Table 1: State-of-the-art results on VQA-CP v2 `test`. All reported models use the same features from [15]. Models with * have been trained by [25]. Models with ** have been trained by [45].

| Model | Overall | Answer type | | |
| --- | --- | --- | --- | --- |
| | | Yes/No | Number | Other |
| Question-Only [10] | 15.95 | 35.09 | 11.63 | 7.11 |
| UpDn [15] ** | 38.01 | . | . | . |
| RAMEN [45] | 39.21 | . | . | . |
| BAN [19] ** | 39.31 | . | . | . |
| MuRel [16] | 39.54 | 42.85 | 13.17 | 45.04 |
| UpDn [15] * | 39.74 | 42.27 | 11.93 | **46.05** |
| UpDn + Q-Adv + DoE [25] | 41.17 | 65.49 | 15.48 | 35.48 |
| Balanced Sampling | 40.38 | 57.99 | 10.07 | 39.23 |
| Q-type Balanced Sampling | 42.11 | 61.55 | 11.26 | 40.39 |
| Baseline architecture (ours) | $38.46 \pm 0.07$ | $42.85 \pm 0.18$ | $12.81 \pm 0.20$ | $43.20 \pm 0.15$ |
| RUBi (ours) | $\mathbf{47.11} \pm 0.51$ | $\mathbf{68.65} \pm 1.16$ | $\mathbf{20.28} \pm 0.90$ | $43.18 \pm 0.43$ |

Table 2: Effectiveness of the RUBi learning strategy when used on different architectures on VQA-CP v2 `test`. Detailed results can be found in the supplementary materials.

| SAN | Overall | UpDn | Overall |
| --- | --- | --- | --- |
| Baseline [26] | 24.96 | Baseline [15] | 39.74 |
| + Q-Adv + DoE [25] | 33.29 | + Q-Adv + DoE [25] | 41.17 |
| + RUBi (ours) | 37.63 | + RUBi (ours) | **44.23** |

Table 3: Overall accuracy of the RUBi learning strategy on VQA v2 `val` and `test-dev` splits.

| Model | val | test-dev |
| --- | --- | --- |
| Baseline (ours) | **63.10** | **64.75** |
| RUBi (ours) | 61.16 | 63.18 |

of 47.11% with a low standard deviation of $\pm 0.51$. This accuracy corresponds to a gain of +5.94 percentage points over the current state-of-the-art UpDn + Q-Adv + DoE. It also corresponds to a gain of +15.88 over GVQA [10], which is a specific architecture designed for VQA-CP. RUBi reaches a +8.65 improvement over our baseline model trained with the classical cross-entropy. In comparison, the second best approach UpDn + Q-Adv + DoE only achieves a +1.43 gain in overall accuracy over their baseline UpDn. In addition, our approach does not significantly reduce the accuracy over our baseline for the answer type *Other*, while the second best approach reduces it by 10.57 point.

**Additional baselines**   We compare our results to two sampling-based training methods. In the *Balanced Sampling* method, we sample the questions such that the answer distribution is uniform. In the *Question-Type Balanced Sampling* method, we sample the questions such that for every question type, the answer distribution is uniform, but the question type distribution remains the same overall Both methods are tested with our baseline architecture. We can see that the *Question-Type Balanced Sampling* improves the result from 38.46 in accuracy to 42.11. This gain is already +0.94 higher than the previous state of the art method [25], but remains significantly lower than our proposed method.

**Architecture agnostic**   RUBi can be used on existing VQA models without changing the underlying architecture. In Table 2, we experimentally demonstrate the generality and effectiveness of our learning scheme by showing results on two additional architectures, Stacked Attention Networks (SAN) [26] and Bottom-Up and Top-Down Attention (UpDn) [15]. First, we show that applying RUBi on these architectures leads to important gains over the baselines trained with their original learning strategy. We report a gain of +11.73 accuracy point for SAN and +4.5 for UpDn. This lower gap in accuracy may show that UpDn is less driven by biases than SAN. This is consistent with results from [25]. Secondly, we show that these architectures trained with RUBi obtain better accuracy than with the state-of-the-art strategy from [25]. We report a gain of +3.4 with SAN + RUBi over SAN + Q-Adv + DoE, and +3.06 with UpDn + RUBi over UpDn + Q-Adv + DoE. Full results splitted by question type are available in the supplementary materials.

**Impact on VQA v2**    We report the impact of our method on the standard VQA v2 dataset in Table 3. VQA v2 train, val and test sets follow the same distribution, contrarily to VQA-CP v2 train and test sets. In this context, we usually observe a drop in accuracy using approaches focused on reducing biases. This is due to the fact that exploiting unwanted correlations from the VQA v2 train set is not discouraged and often leads to a higher accuracy on the test set. Nevertheless, our RUBi approach leads to a comparable drop to what can be seen in the state-of-the-art. We report a drop of 1.94 percentage points with respect to our baseline, while [10] report a drop of 3.78 between GVQA and their SAN baseline. [25] report drops of 0.05, 0.73 and 2.95 for their three learning strategies with the UpDn architecture which uses the same visual features as RUBi. As shown in this section, RUBi improves the accuracy on VQA-CP v2 from a large margin, while maintaining competitive performance on the standard VQA v2 dataset compared to similar approaches.

**Validation of the masking strategy**    We compare different fusion techniques to combine the output of $nn_q$ with the output from the VQA model. We report a drop of 7.09 accuracy point on VQA-CP v2 by replacing the sigmoid with a ReLU on our best scoring model. Using an element-wise sum instead of an element-wise product leads to a further performance drop. These results confirm the effectiveness of our proposed masking method which relies on a sigmoid and an element-wise sum.

**Validation of the question-only loss**    In Table 4, we validate the ability of the question-only loss $\mathcal{L}_{QO}$ to reduce the question biases. The absence of $\mathcal{L}_{QO}$ implies that the question-only classifier $c_q$ is never used, and $nn_q$ only receives gradients from the main loss $\mathcal{L}_{QM}$. Using $\mathcal{L}_{QO}$ leads to consistent gains on all three architectures. We report a gain of +0.89 for our Baseline architecture, +0.22 for SAN, +4.76 for UpDn.

| Model | $\mathcal{L}_{QO}$ | Overall | Yes/No | Number | Other |
|---|---|---|---|---|---|
| Baseline + RUBi | ✓ | **47.11** | 68.65 | 20.28 | **43.18** |
|  | ✗ | 46.11 | **69.18** | **26.85** | 39.31 |
| SAN + RUBi | ✓ | **37.63** | 59.49 | **13.71** | **32.74** |
|  | ✗ | 36.96 | **59.7**8 | 12.55 | 31.69 |
| UpDn + RUBi | ✓ | **44.23** | **67.05** | **17.48** | **39.61** |
|  | ✗ | 39.47 | 60.27 | 16.01 | 35.01 |

Table 4: Ablation study of the question-only loss $\mathcal{L}_{QO}$ on VQA-CP v2.

## 4.2   Qualitative analysis

To better understand the impact of our RUBi approach, we compare in Figure 4 the answer distribution on VQA-CP v2 for some specific question patterns. We also display interesting behaviors on some examples using attention maps extracted as in [16]. In the first row, we show the ability of RUBi to reduce biases for the *is this person skiing* question pattern. Most examples in the train set have the answer *yes*, while in the test set, they have the answer *no*. Nevertheless, RUBi outputs 80% of *no*, while the baseline almost always outputs *yes*. Interestingly, the best scoring region from the attention map of both models is localized on the shoes. To get the answer right, RUBi seems to reason about the absence of skis in this region. It seems that our baseline gets it wrong by not seeing that the skis are not locked under the ski boots. This unwanted behavior could be due to the question biases. In the second row, similar behaviors occur for the *what color are the bananas* question pattern. 80% of the answers from the train set are *yellow*, while most of them are *green* in the test set. We show that the amount of *green* and *white* answers from RUBi are much closer to the ones from the test set than with our baseline. In the example, it seems that RUBi relies on the color of the banana, while our baseline misses it. In the third row, it seems that RUBi is able to ground the textual concepts such as *top part of the fire hydrant* and *color* on the right visual region, while the baseline relies on the correlations between the fire hydrant, the yellow color of its core and the answer *yellow*. Similarly on the fourth row, RUBi grounds *color*, *star*, *fire hydrant* on the right region, while our baseline relies on correlations between *color*, *fire hydrant*, the yellow color of the top part region and the answer *yellow*. Interestingly, there is no similar question that involves the color of a star on a fire hydrant in the training set. It shows the capacity of RUBi to generalize to unseen examples by composing and grounding existing visual and textual concepts from other kinds of question patterns.

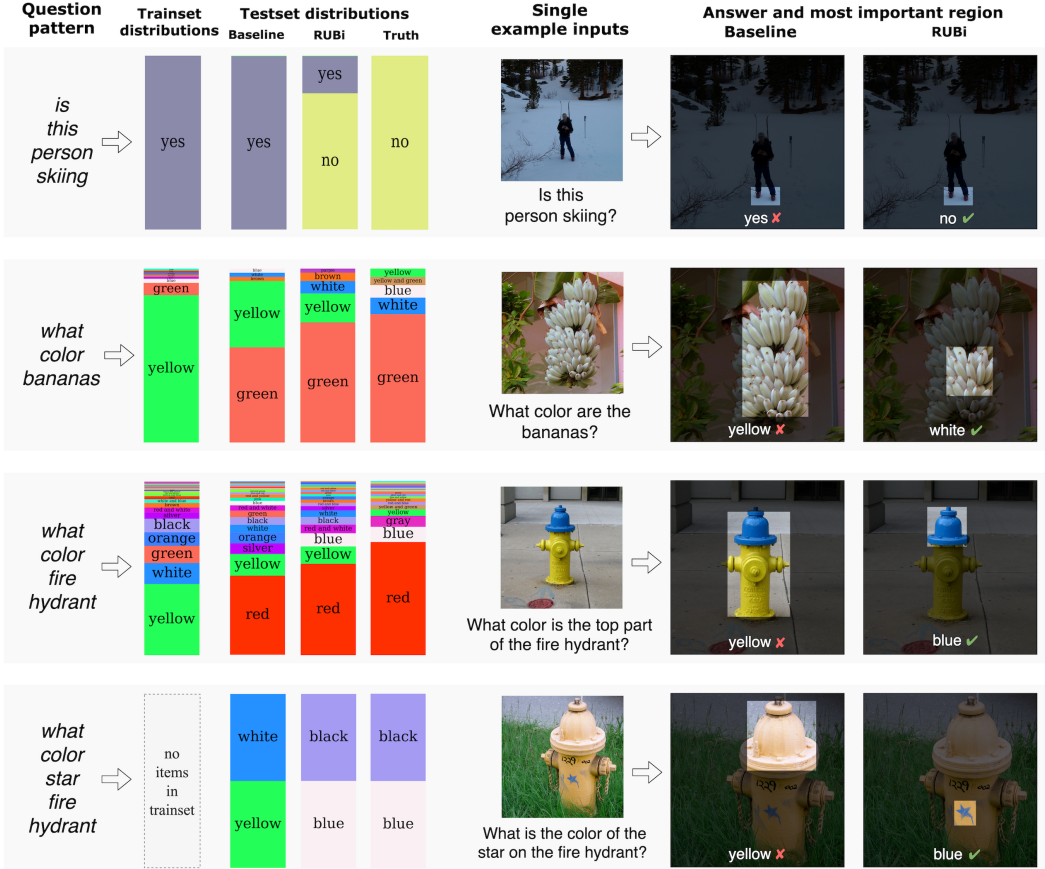

Figure 4: Qualitative comparison between the outputs of RUBi and our baseline on VQA-CP v2 `test`. On the left, we display distributions of answers for the train set, the baseline evaluated on the test set, RUBi on the test set and the ground truth answers from the test set. For each row, we filter questions in a certain way. In the first row, we keep the questions that exactly match the string *is this person skiing*. In the three other rows, we filter questions that respectively include the following words: *what color bananas*, *what color fire hydrant* and *what color star hydrant*. On the right, we display examples that contains the pattern from the left. For each example, we display the answer of our baseline and RUBi, as well as the best scoring region from their attention map.

## 5 Conclusion

We propose RUBi to reduce unimodal biases learned by Visual Question Answering (VQA) models. RUBi is a simple learning strategy designed to be model agnostic. It is based on a question-only branch that captures unwanted statistical regularities from the question modality. This branch influences the base VQA model to prevent the learning of unimodal biases from the question. We demonstrate a significant gain of +5.94 percentage point in accuracy over the state-of-the-art result on VQA-CP v2, a dataset specifically designed to account for question biases. We also show that RUBi is effective with different kinds of common VQA models. In future works, we would like to extend our approach on other multimodal tasks.

## Acknowledgments

We would like to thank the reviewers for valuable and constructive comments and suggestions. We additionally would like to thank Abhishek Das and Aishwarya Agrawal for their help.

The effort from Sorbonne University was supported within the Labex SMART supported by French state funds managed by the ANR within the Investissements d'Avenir programme under reference ANR-11-LABX-65.

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
