[Supplementary Material]

# 6 Supplementary materials

Figure 5: Visual comparison between RUBi and [25].

## 6.1 Additional experiments

**Results on VQA-CP v1** In Table 5, we report results on the VQA-CP v1 dataset [10]. Our RUBi approach consistently leads to significant gains over the classical learning strategy with a gain of +9.8 overall accuracy point with our baseline architecture, +19.2 with SAN and +7.66 with UpDn. Additionally, RUBi leads to a gain of +2.65 over the adversarial regularization method (AdvReg) from [25] with SAN. A visual comparison between RUBi and [25] can be found in Figure 5. Finally, all three architectures trained with RUBi reach a higher accuracy than GVQA [10] which has been hand-designed to overcome biases.

| Model | Overall | Yes/No | Number | Other |
|---|---|---|---|---|
| GVQA [10] | 39.23 | 64.72 | 11.87 | 24.86 |
| Baseline (ours) | 37.13 | 41.96 | 12.54 | 41.35 |
| Baseline + RUBi | **46.93** | **66.78** | **20.98** | **43.64** |
| SAN [25] | 26.88 | 35.34 | 11.34 | 24.70 |
| SAN + AdvReg [25] | 43.43 | 74.16 | 12.44 | 25.32 |
| SAN + RUBi | **46.08** | **75.00** | **13.30** | **30.49** |
| UpDn (ours) | 37.15 | 41.13 | 12.73 | **43.00** |
| UpDn + RUBi | **44.81** | **69.65** | **14.91** | 32.13 |

Table 5: Overall accuracy top1 on VQA-CP v1

**Detailed results on VQA-CP v2** In Table 6, we report the full results of our experiments for SAN and UpDn architectures on the VQA-CP v2 dataset.

**Quantitative study of the grounding ability on VQA-HAT** We conduct additional studies to evaluate the grounding ability of models trained with RUBi. We follow the experimental protocol of VQA-HAT [44]. We train our models on VQA v1 train set and evaluate them using rank-correlation on the VQA-HAT val set, which is a subset of the VQA v1 val set. This metric compares attention maps computed from a model against human annotations indicating which regions humans found relevant for answering the question. In Table 7, we report a gain of +0.012 with our baseline architecture trained with RUBi, a gain of +0.019 with SAN and a loss of -0.003 with UpDn architecture. In future

| Model | Overall | Yes/No | Number | Other |
|---|---|---|---|---|
| SAN [26] | 24.96 | 38.35 | 11.14 | 21.74 |
| SAN + RUBi | **37.63** | **59.49** | **13.71** | **32.74** |
| UpDn [15] | 39.74 | 42.27 | 11.93 | **46.05** |
| UpDn + RUBi | **44.23** | **67.05** | **17.48** | 39.61 |

Table 6: Overall accuracy top1 on VQA-CP v2 for the SAN and UpDn architectures.

works, we would like to go beyond these early results in order to further evaluate the impact on grounding induced by RUBi.

| Model | RUBi | Rank-Corr. |
|---|---|---|
| Random [44] | | 0.000 |
| Human [44] | | 0.623 |
| Baseline | ✗ | 0.431 |
| | ✓ | **0.443** |
| SAN | ✗ | 0.191 |
| | ✓ | **0.210** |
| UpDn | ✗ | **0.449** |
| | ✓ | 0.446 |

Table 7: Correlation with Human Attention Maps on VQA-HAT `val` set [44].

**Qualitative study of the grounding ability on VQA-HAT**  We display in Figure 6 and Figure 7 some manually selected VQA triplets associated to the human attention maps provided by VQA-HAT [44] and the attention maps computed from our baseline architecture when trained with and without RUBi. In Figure 6, we observe that the attention maps with RUBi are closer to the human attention maps than without RUBi. On the contrary, we observe in Figure 7 some failure to improve grounding ability.

Is the border sporting a goatee? yes     Human Attention     Baseline Attention     Baseline + RUBi Attention

Is the fire hydrant inside the fence? no     Human Attention     Baseline Attention     Baseline + RUBi Attention

What flavors is the frosting on the donut? chocolate     Human Attention     Baseline Attention     Baseline + RUBi Attention

What type of vehicle is parked? car     Human Attention     Baseline Attention     Baseline + RUBi Attention

What is the white object behind the cat? ant trap     Human Attention     Baseline Attention     Baseline + RUBi Attention

Figure 6: Examples of better grounding ability on VQA-HAT implied by RUBi. From the left column to the right: image-question-answer triplet, human attention map from [44], attention map from our baseline, attention map from our baseline trained with RUBi.

Figure 7: Examples of failure to improve grounding ability on VQA-HAT. From the left column to the right: image-question-answer triplet, human attention map from [44], attention map from our baseline, attention map from our baseline trained with RUBi.

## 6.2 Implementation details

**Image encoder**    We use the pretrained Faster R-CNN by [15] to extract object features. We use the setup that extracts 36 regions for each image. We do not fine-tune the image extractor.

**Question encoder**    We use the same preprocessing as in [16]. We apply a lower case transformation and remove the punctuation. We only consider the most frequent 3000 answers for both VQA v2 and VQA CP v2. We then use a pretrained Skip-thought encoder with a two-glimpses self attention mechanism. The final embedding is of size 4800. We fine-tune the question encoder during training.

**Baseline architecture**    Our baseline architecture is a simplified version of the MuRel architecture [16]. First, it computes a bilinear fusion between the question vector and the visual features for each region. The bilinear fusion module is a BLOCK [17] composed of 15 chunks, each of rank 15. The dimension of the projection space is 1000, and the output dimension is 2048. The output of the bilinear fusion is aggregated using a max pooling over $n_v$ regions. The resulting vector is then fed into a MLP classifier composed of three layers of size (2048, 2048, 3000), with ReLU activations. It outputs the predictions over the space of the 3000 answers.

**Question-only branch**    The RUBi question-only branch feeds the question into a first MLP composed of three layers, of size (2048, 2048, 3000), with ReLU activations. First, this output vector goes through a sigmoid to compute the mask that will alter the predictions of the VQA model. Secondly, this same output vector goes through a single linear layer of size 3000. We use these question-only predictions to compute the question-only loss.

**Optimization process**    We train all our models with the Adam optimizer. We train our baseline architecture with the learning rate scheduler of [16]. We use a learning rate of $1.5 \times 10^{-4}$ and a batch size of 256. During the first 7 epochs, we linearly increase the learning rate to $6 \times 10^{-4}$. After epoch 14, we apply a learning rate decay strategy which multiplies the learning rate by 0.25 every two epochs. We train our models until convergence as we do not have a validation set for VQA-CP v2. For the UpDn and SAN architectures, we follow the optimization procedure described in [25].

**Software and hardware**    We use pytorch 1.1.0 to implement our algorithms in order to benefit from the GPU acceleration. We use four NVidia Titan Xp GPU in this study. We use a single GPU for each experiments. We use a dedicated SSD to load the visual features using multiple threads. A single experiment from Table 1 with the baseline architecture trained with or without RUBi takes less than five hours to run.