[Reviews · NeurIPS 2019]

Reviewer 1



Originality: The proposed method is a novel dynamic loss re-weighting technique applied to VQA under changing priors condition, aka VQA-CP, where the train and test sets are deliberately constructed to have different distributions. The related works are adequately cited and discussed. While prior works have also focused on using knowledge from a question-only model to capture unnecessary biases in the dataset [25], the paper differs from [25] in some key aspects. E.g., the proposed model guides the whole model (including the visual encoding branch) to learn "harder" examples better whereas [25] focuses on only reducing bias from question encoding. Quality: The proposed method is sound and well-motivated. The experimental setup is mostly sound and is on par with prior works. I have some qualms about absence of some common-sense baselines but that is not entirely the authors fault since the prior works have also failed to set a precedent. My another minor issue with the paper is lack of discussion of shortcomings / future work for the paper. Clarity: The paper is very easy to read and the major contributions and background work is clearly laid out. The method descriptions are complete and contain enough detail to be able to reproduce. Significance: The paper overall has a moderate significance. The proposed method is sound and is clearly demonstrated to work well for existing dataset. It is likely that the proposed method will be used in other bimodal tasks as well. However, common with existing works in this space, the algorithm is designed with prior knowledge about how exactly the (artificially perturbed) VQA-CP was constructed. Therefore, to me, it is slightly less significant compared to (hypothetical) alternative algorithms that propose a robust vision-language understanding rather than developing specialize algorithms just for "reducing" effects of bias on an specially constructed dataset. It is still a valuable contribution, just a very specific one instead of a true general solution. *** POST REBUTTAL COMMENTS *** As I said during my original review, I think that the "good results" obtained on VQA-CP alone is not *very* significant as there haven't really been carefully established baselines. I thank the authors for their rebuttal, which already shows that even the most naive baselines perform over 4% over the existing numbers. Again, I do not put the burden of this on the authors; but this is something the whole community has sort of ignored. That being said, the improvements to the robustness by the proposed model is undeniable. In the rebuttal the authors also show that it improves robustness in VQA-HAT. Most importantly, the proposed method is an model-and-task agnostic de-biasing technique that I think will be useful to present to large community in NeurIPS. After reading other reviews and author rebuttal, I am raising my score to a 7.

Reviewer 2



The paper proposed a strategy which guides the model logits toward more ‘question biased’ logits. The effect of this is that when the prediction is the same as the label, the loss is smaller. If the prediction is not the same, then the loss is larger.  The model achieves this by merging the normal VQA model logits with the logits that is generated based on question only. During testing, the path from the question-only side is removed so that only the VQA logits will be used during prediction.  The experiments show the method outperforms existing methods in the VQA CP data.  The paper is well written and easy to follow.  The role of the c’_q classifier is a bit unclear. Since the outputs of c_q are already the logits, why is it necessary to feed c_q logits into another classifier? What if the logits of c’_q and c_q do not match each other? Is there any way to enforce the consistency of the outputs of c’_q and c_q?  The way of combining question-only model is not unique. Why choosing this model? Have you tried any other candidates?  The method seems to perform well in the VQA CP data. However, its use case may be very limited. It is only applicable in VQA where question biases are present. It cannot be adapted to other applications. It also does not address other robustness issues of VQA, for example, the circle consistency. Therefore, although this idea seems novel and valid for this application, its impact may be low. *** After reading the author feedback *** I appreciate the additional experiments and clarifications wrt the role of c'_q and the ablation wrt the combination of the q-model. This clarifications and the additional results add strength to the paper.

Reviewer 3



Summary - The authors address the task of VQA with a specific interest in minimizing unimodal bias (specifically, language bias). For that they propose a general approach that can be combined with different VQA architectures. The main idea is to make use of a question-only model to VQA (no image input), and reduce/increase the loss of the data points that are correctly/incorrectly answered by the question-only model. - The experiments are carried out on the VQA-CP v2 dataset. A direct comparison with the prior state-of-the-art [25] (with the same underlying VQA architecture) shows a superior performance of the proposed method. The authors also show that their method only leads to a small drop in performance on the standard VQA v2 dataset. - The qualitative results in Fig 4 are interesting and informative. Originality - The proposed is overall novel, to the best of my knowledge. Nevertheless, it resembles the method of [25], which while mentioned, could be compared to the proposed approach more thoroughly. E.g. [25] also contains a question-only branch to “unbias” the VQA models. It would help if the authors illustrate the two approaches side by side in Fig 2 and provide a detailed comparison. - There are other recent works exposing unimodal bias in tasks such as embodied QA [A,B], vision-and-language navigation [B] and image captioning [C]. [A] Ankesh Anand, Eugene Belilovsky, Kyle Kastnerand, Hugo Larochelle, and Aaron Courville. Blindfold baselines for embodied QA. NeurIPS 2018 Workshop on Visually Grounded Interaction and Language (ViGIL). [B] Jesse Thomason, Daniel Gordon, and Yonatan Bisk. Shifting the baseline: Single modality performance on visual navigation & QA. NAACL 2019. [C] Anna Rohrbach, Lisa Anne Hendricks, Kaylee Burns, Trevor Darrell, and Kate Saenko. Object hallucination in image captioning. EMNLP 2018. Quality - It is not entirely clear how the classifier c_q is trained; L150 mentions a cross-entropy loss, but it does not seem to be visualized in the approach overview Fig 2? (I assume that the classifier c_q represents the question-only VQA model?) - Later another loss, L_QO, is introduced in L184 for the classifier c_q’; it is supposed to “further improve the unimodal branch ability to capture biases”; what is the connection of this loss to the one mentioned above? What is the role of the classifier c_q’? - The experiments are only carried out on VQA-CP v2, not on VQA-CP, as done in [25]. - There is no discussion of why the proposed baseline benefits from the proposed approach much more than e.g. the UpDn model (Table 1 vs. Table 2). - It would be informative to include the detailed evaluation breakdown (by answer type) for the experiments in Table 2 (similar to Table 1), for a more complete “apples-to-apples” comparison to [25]. Clarity - The authors title the paper/approach “Reducing Unimodal Biases”, while they specifically focus on language biases (question/answer statistics). Some prior work has analyzed unimodal biases, considering both language-only and vision-only baselines, e.g. [B]. It seems more appropriate to say “language biases” here. - It is not clear from Fig 2 where the backpropagation is or is not happening (it becomes more clear from Fig 3). - L215 makes a reference to GVQA [10] performance, but it is not included in Table 1. - The prior approaches in Table 1 are not introduced/discussed. - Writing: L61: from => by, L73: about => the, L196 merge => merges Significance - There is a great need to address bias in deep learning models, in particular in vision-and-language domain, making this work rather relevant. While the proposed method does not appear groundbreakingly novel, it is nevertheless quite effective. The main issues of the submission are listed above (e.g. comparison to [25], confusing parts in the approach description, somewhat limited evaluation). Other The authors have checked "Yes" for all questions in the Reproducibility checklist, although not all are even relevant. UPDATE I appreciate the new baselines/ablations that the authors provided upon R2’s and R3’s request. Besides, they included convincing results on VQA-CP v1 and promise to address some of my other concerns in the final version. I thus keep my original score “7”.

[Author Response · NeurIPS 2019]

We thank the reviewers for their useful comments and suggestions. We are glad that the reviewers found our approach to be novel (R2, R3, R4), general and significant (R4), a valuable contribution (R2), appreciated its superior performance (R2, R3, R4), and found our paper to be clear (R2, R3). We now address their requests and concerns.

**Answers to R2:**

- **Q1 Additional baselines:** We ran this baseline of sampling answers from a uniform distribution. This gets an accuracy of 40.25% (compared to 47.11% with our approach using the same baseline architecture). As a recall, our current baseline gets 38.46%. Inspired by this suggestion, we also tested sampling answers from a uniform distribution per question-type. This gets an accuracy of 42.11%. We will add these two new baselines in Table 1.

- **Q2 Grounding ability, interpretability and future works:** We ran new experiments on the VQA-HAT dataset to quantitatively validate that models trained with the RUBi strategy on VQA 1.0 improves the ability to *attend to the "right" regions of the image*. We report 0.4551 in rank-correlation (higher is better) with our baseline architecture and 0.4671 when trained with RUBi (see Table 2 in VQA-HAT paper for reference; recall that we use image features from [15]). Interestingly, our approach improves the grounding ability without being designed to do so explicitly. We will add a new table of results on VQA-HAT including different architectures, as well as qualitative results similar to the attention maps from Figure 6 of the VQA-HAT paper. These visualizations will allow us to discuss about interpretability and grounded/symbolic reasoning. Also, we will add details about future works in the conclusion.

**Answers to R3:**

- **Q1 Significance of $c'_q$:** We ran new experiments to evaluate the usefulness of $c'_q$. First, we fixed $c'_q$ to be the identity (i.e. we removed $c'_q$ while $c_q$ receives gradients from $L_{QO}$). We report an accuracy of 5.38% on VQA-CP v2 with our baseline architecture. This low performance is expected since $c_q$ is designed to output a 0-1 mask using the sigmoid, and not to output logits. We agree that the term "classifier" to define $c_q$ was unclear. We will change it. Secondly, we removed both $c'_q$ and the question-only loss $L_{QO}$. We report a slightly lower accuracy of 46.08% (-1.03 compared to a training with the full RUBi strategy) for the baseline architecture. Intuitively, the 0-1 masks produced by $c_q$ must be good enough to reduce the importance of biases early during training. $c'_q$ and $L_{QO}$ provides an additional supervision to $c_q$ helping it to generate better masks, earlier in the training. We will add a new table of results about $c'_q$. We will also improve the discussion about $c'_q$ and $L_{QO}$.

- **Q2 Comparison with other candidate models:** We experimented with different fusion techniques to combine the output of $c_q$ with the output from the VQA model. For instance, a ReLU instead of a sigmoid gets 40.02% (compared to 47.11% with our approach using the same baseline architecture). Other classical fusions such as an element-wise sum lead to more significant performance drop than what was previously reported with ReLU. Upon acceptance, we will add a detailed discussion about these fusions in the final paper.

**Answers to R4:**

- **Q1 Visual comparison to [25]:** We will add to Figure 2 an *"apples-to-apples" comparison to [25]* as depicted in the figure of this rebuttal. Similarly to the "gradient negation" illustration, we will improve Figure 2 to indicate *when the backpropagation is not happening* in $e_q$. We will also clarify the comparison with [25], from line 113 to 122.

- **Q2 Clarification about $c_q$ and $c'_q$:** We will clarify that $c_q$ receives gradients from $L_{QM}$ and $L_{QO}$. See the answer Q1 to R3 for further information about $c_q$ and $c'_q$.

- **Q3 Evaluation on VQA-CP v1 and detailed evaluation breakdown:** We ran new experiments on VQA-CP v1 and report state-of-the-art results regardless of the architecture trained with RUBi. Our approach consistently leads to significant gains over the classical learning strategy. We report improvements of +9.80 in overall accuracy with our baseline architecture, +10.46 with UpDn, +19.23 with SAN. We will add a new table of results on VQA-CP v1 similarly to Table 1. We will also include the accuracy for each answer types for the UpDn and SAN architectures in Table 2.

- **Q4 Discussion about [A,B,C] and prior approaches:** We will add [A,B,C] to the related works section to highlight the importance of biases reducing methods in the multimodal context. Finally, we will introduce [15,41,19,16] from Table 1 in the state-of-the-art comparison paragraph. Note that these previous approaches do not focus on biases reduction contrary to [25].

| Model | Overall |
|---|---|
| GVQA [10] | 39.23 |
| SAN [26] | 26.88 |
| + [25] | 43.43 |
| + RUBi | **46.11** |
| UpDn [15] | 37.15 |
| + RUBi | **47.61** |
| Baseline | 37.13 |
| + RUBi | **46.93** |

[Meta-Review · NeurIPS 2019]

After the authors' rebuttal all reviewers believe the paper makes a significant enough contribution to be accepted to the conference. When there is a need to obtain large amounts of data for complex tasks such as VQA, bias in the labeling process is highly likely. Techniques that improve robustness to such biases can have a significant impact in these cases. The authors should incorporate the clarifications and results from the rebuttal into the paper and address the reviewers comments.